# On the Impact of Cross-Domain Data on German Language Models

**Amin Dada[1], Aokun Chen[2,3], Cheng Peng[2,3], Kaleb E Smith[4], Ahmad Idrissi-Yaghir[5,6],
Constantin Marc Seibold[1,7], Jianning Li[1], Lars Heiliger[1], Christoph M. Friedrich[5,6],
Daniel Truhn[8], Jan Egger[1,9], Jiang Bian[2,3], Jens Kleesiek[1,9,10,11], Yonghui Wu[2,3]**

[1]Institute for AI in Medicine (IKIM), University Hospital Essen (AöR), Essen, Germany

[2]Department of Health Outcomes and Biomedical Informatics, College of Medicine, University of Florida, Gainesville, FL, USA

[3]Cancer Informatics and eHealth core, University of Florida Health Cancer Center, Gainesville, FL, USA

[4]NVIDIA, Santa Clara, CA, USA

[5]Department of Computer Science, University of Applied Sciences and Arts Dortmund, Dortmund, Germany

[6]Institute for Medical Informatics, Biometry and Epidemiology (IMIBE), University Hospital Essen (AöR), Essen, Germany

[7]Clinic for Nuclear Medicine, University Hospital Essen (AöR), Essen, Germany

[8]Department of Diagnostic and Interventional Radiology, University Hospital RWTH Aachen, Aachen, Germany

[9]Cancer Research Center Cologne Essen (CCCE), West German Cancer Center Essen,

University Hospital Essen (AöR), Essen, Germany

[10]German Cancer Consortium (DKTK, Partner site Essen), Heidelberg, Germany

[11]Department of Physics, TU Dortmund, Dortmund, Germany

## Abstract

Traditionally, large language models have been either trained on general web crawls or domain-specific data. However, recent successes of generative large language models, have shed light on the benefits of cross-domain datasets. To examine the significance of prioritizing data diversity over quality, we present a German dataset comprising texts from five domains, along with another dataset aimed at containing high-quality data. Through training a series of models ranging between 122M and 750M parameters on both datasets, we conduct a comprehensive benchmark on multiple downstream tasks. Our findings demonstrate that the models trained on the cross-domain dataset outperform those trained on quality data alone, leading to improvements up to $4.45\%$ over the previous state-of-the-art. The models are available at: https://huggingface.co/ikim-uk-essen

## 1 Introduction

With established scaling laws (Kaplan et al., 2020), increasing the model size and datasets has become a consistent scheme in the rapidly evolving field of Large Language Models (LLMs) (Kaplan et al., 2020; Touvron et al., 2023; Smith et al., 2022). This pattern exists for any recently proposed architecture (Yang et al., 2023), independent of decoder-only (Brown et al., 2020b; Chowdhery et al., 2022; Touvron et al., 2023), encoder-only (He et al., 2021; Devlin et al., 2019; Liu et al., 2019; Clark et al.,

2020), or encoder-decoder models (Raffel et al., 2020; Zeng et al., 2022). As these have shown impressive performance for natural language understanding, many works built upon them for specific tasks such as medical language understanding (Rasmy et al., 2021; Lee et al., 2020) or human interaction (Bai et al., 2022), or advance them through intelligent design choices. For instance, He et al. (2021) introduced the disentangled attention mechanism that encodes the token position and information separately, enabling it to surpass human performance on the SuperGLUE benchmark (Wang et al., 2019). However, as the data requirements for LLMs can become difficult to acquire for smaller institutions, it poses the questions whether strictly increasing the data amount is the only solution or if it is possible to enable the training of LLMs through a more elaborate data selection process.

The Pile (Gao et al., 2020) has played a crucial role in emphasizing the advantages of augmenting web-crawl datasets with a wide array of domain-specific data. This realization has prompted the training of large language models (LLMs) on The Pile and similar datasets (Gao et al., 2020; Laurençon et al., 2022). Gunasekar et al. (2023) presented a 1.3B parameter model trained on a dataset of 7B carefully curated tokens that is comparable to GPT3.5 on HumanEval indicating that data quality apart from general scaling laws (Kaplan et al., 2020) play an important role as well.

However, a notable disparity arises when considering the German language, as there is currently a lack of comparable variety-focused datasets. To address this gap, we propose various methods for curating a diverse German dataset, even for domains with limited resources, such as the medical field. Our methods are applicable to a wide range of languages, as approximately 74% of this dataset is acquired through automatic translation and web-crawl filtering techniques. We demonstrate the advantages of such datasets over pure web-crawl-based datasets with a set of encoder-only models we call GeBERTa.

In 2020, Chan et al. published a set of German transformer (Vaswani et al., 2017) models based on BERT and ELECTRA, achieving state-of-the-art results on two downstream datasets. Around the same time, GottBERT was released, a German RoBERTa model (Scheible et al., 2020). Since the two publications were only evaluated on two common downstream datasets and both dispensed with hyper-parameter tuning, it is unclear how they compare. In this work, we train new German LMs with the DeBERTa (He et al., 2021) architecture and assemble a comprehensive benchmark of eight downstream datasets. By fine-tuning the hyper-parameters of our models and the previously released ones, we showcase the enhanced model performance achieved through the DeBERTa architecture and our cross-domain dataset. (Chan et al., 2020) and (Scheible et al., 2020) were based on the web-based OSCAR dataset. Moreover, Chan et al., 2020 explored the impact of data quantity on model performance. The aim of this work is to quantify the effects of data quality and the inclusion of cross-domain data on encoder-only models.

Our contributions can be summarized as follows:

- We empirically show that pre-training language models on heterogenous datasets results in better downstream performance.

- We assemble an extensive benchmark of eight tasks and thoroughly evaluate previously published models.

- We present new German language models that achieve state-of-the-art results.

- We release our models and the source code for compiling the pre-training dataset.

## 2 Related Work

### 2.1 German LMs

Subsequent to the immense success achieved by English models, a surge of transformer models trained in various other languages has emerged (Martin et al., 2020; Chan et al., 2020). This development has been made possible through the availability and utilization of web-based multilingual corpora. Most prominently OSCAR (Ortiz Suarez and Gabay, 2022) and CC100 (Wenzek et al., 2020). While Oscar only performs some heuristics to filter quality data, the CC100 pipeline makes a deliberate attempt to filter for higher-quality texts by using an n-gram model to filter for texts that are similar to Wikipedia articles (Wenzek et al., 2020). We base our study on GC4[1] a dataset that was collected with the CC100 pipeline. In German, (Chan et al., 2020) released GBERT$_{Base}$, GBERT$_{Large}$, GELECTRA$_{Base}$, and GELECTRA$_{Large}$, which were trained on OSCAR, Wikipedia, the Open Parallel Corpus (Tiedemann, 2012), and legal texts (Ostendorff et al., 2020a). In their study, Chan et al. explored the impact of data quantity on model performance. Their dataset consisted of a combination of legal documents and a small subset of medical documents from the Open Parallel Corpus, which introduced a cross-domain aspect to their data. However, they did not explicitly measure the specific effects of incorporating cross-domain data. In addition, GottBERT (Scheible et al., 2020) was released, a German RoBERTa (Liu et al., 2019) model that was solely trained on the German portion of the OSCAR dataset.

### 2.2 Cross-Domain Pre-training

Encoder-only transformers are usually initially trained on general web crawls, news articles, books, and subsequently fine-tuned for specific domains. A well-known example is BioBERT (Lee et al., 2020), a domain-specific LM further pre-trained on PubMed texts based on BERT (Devlin et al., 2019), which is pre-trained on Wikipedia and BooksCorpus. Alsentzer et al., 2019, however, show that pre-training on medical data (Clinical BioBERT) is superior to pre-training on general data (Clinical BERT) for several downstream medical tasks. Alternatively, there are some works that only train on the target domain, for instance, (Gu et al., 2021)

---

[1] https://german-nlp-group.github.io/projects/gc4-corpus.html (last access: 20-06-2023)

and (Yang et al., 2022). On the other hand, generative LLMs are often trained on more diverse datasets that incorporate multiple domains. After the release of GPT-2 (Radford et al., 2019) and GPT-3 (Brown et al., 2020a), the research focus has shifted to diverse large datasets. The Pile combines web data with forum posts, scientific publications, and articles from a large range of domains and demonstrates that data diversity is an important factor for the performance of LLMs. Although the composition of the training data for GPT-3.5 and GPT-4 remains unknown, several publications have demonstrated their capabilities in multiple domain-specific tasks, including medicine (Adams et al., 2023; Nori et al., 2023), implying that the training data incorporates various domains.

The use of cross-domain datasets in other languages remains rare, mainly because of the effort involved in compiling such datasets. In addition, the availability of domain-specific data is low in many languages. For example, a recent study found only less than 2 million publicly available medical documents in German (Bressem et al., 2023). In this study, we present different methods to create corresponding German datasets based on web crawls or English domain-specific datasets. Although, due to the use of automatic translation and filtering methods, the quality is lower compared to the English datasets, we can show that our cross-domain dataset leads to a better performance of German language models.

## 3 Data Collection

CC100 prioritizes quality in its construction. While this emphasis on quality might be beneficial, it also has certain limitations when it comes to the dataset. Specifically, domains that encompass informal or noisy texts, such as social media posts or medical documents, may be at a disadvantage due to the focus on quality. To address this concern and explore the trade-off between quality and variety, we have developed two distinct datasets. The first dataset $\mathcal{D}_{\text{quality}}$ is centered around ensuring high quality, featuring carefully curated and reliable content. However, recognizing the importance of incorporating diverse texts, we have taken steps to extend the quality-focused dataset to include a broader range of material, thereby increasing its variety. By creating these two datasets with different emphases, we aim to examine whether the advantages gained from a variety-focused approach

outweigh any potential drawbacks associated with reduced data quality.

### 3.1 Quality-Focused Dataset

We combine GC4, German news articles, and other relevant resources to create our first dataset $\mathcal{D}_{\text{quality}}$. While (Wenzek et al., 2020) had previously applied deduplication based on document hashes, the creators of GC4 implemented an additional deduplication step. They employed MongoDB deduplication to identify duplicates across snapshots (Wenzek et al., 2020). To construct our dataset, we relied on all pre-processed GC4 head snapshots, which showed the highest similarity to Wikipedia articles. Additionally, we incorporated the WMT 2022 German monolingual news crawl corpus (Kocmi et al., 2022) and the 20220301.de German Wikipedia dump[2].

Given the potential overlap between these datasets, we applied the deduplication algorithm introduced by (Lee et al., 2022). The deduplication process involved two stages: initially, we deduplicated GC4 and the news corpus separately, and subsequently, deduplicated the combined datasets. We configure the minimal matching substring length to 100 tokens. To ensure document integrity, any document containing duplicates was removed entirely.

Interestingly, despite the previous deduplication efforts by (Wenzek et al., 2020), which removed 70% of the texts from each GC4 snapshot, and the subsequent MongoDB deduplication, which eliminated an additional 46% of the remaining data, we discovered that out of an initial 74.6 billion tokens in GC4, 35.3 billion were duplicates. Moreover, out of the 16 billion news tokens, 9.3 billion were identified as duplicates. Conversely, when considering the concatenation of all datasets, only 0.72 billion token duplicates were found.

### 3.2 Variety-Enhanced Dataset

In this section, we aim to expand upon the quality-focused dataset we discussed earlier, with the primary objective of enhancing its diversity. To achieve this, we introduce several additions to $\mathcal{D}_{\text{quality}}$. We partition the data we gather into four distinct domains: informal, medical, legal, and literature. The subsequent section will provide a detailed account of how we obtain these domain-specific datasets. By including data from various

---

[2] https://huggingface.co/datasets/wikipedia (last access: 22-06-2023)

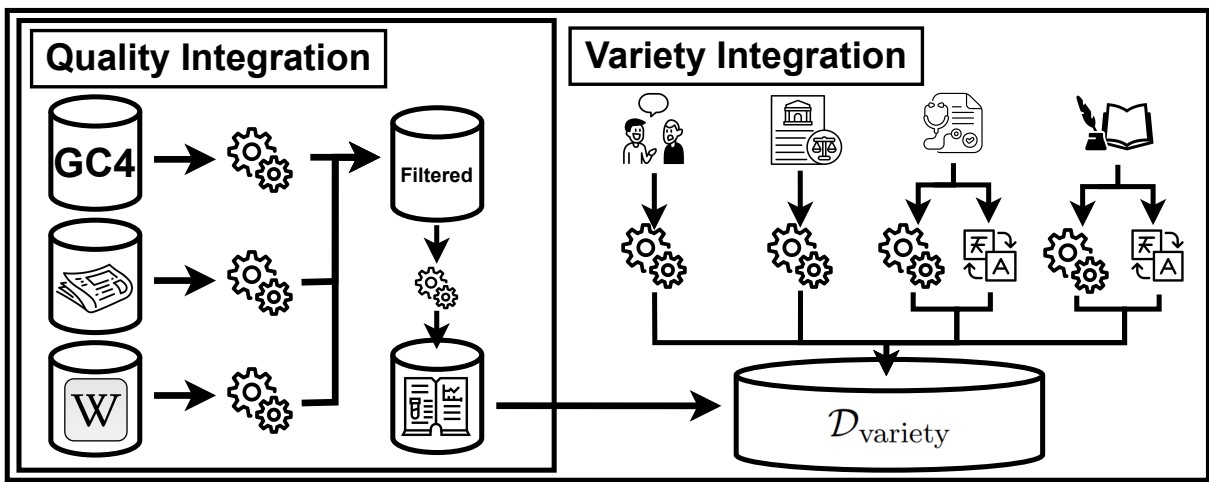

Figure 1: Overview of our data set collection process. The deduplicated quality-focussed data set is based on GC4, news, and Wikipedia. We extend this data set with more variety-focussed data sources that form filtered multilingual data sets, translations, and curated data.

| Category | Source Data | Trans. | Data Size | #Docs | #Tokens |
|---|---|---|---|---|---|
| | Wikipedia | No | 9GB | 2,665,357 | 1.9B |
| | News | No | 28GB | 12,305,326 | 6.1B |
| | GC4 | No | 90GB | 31,669,772 | 19.4B |
| | Reddit 2019-2023 (GER) | No | 5.8GB | 15,036,592 | 1.3B |
| | Holiday Reviews | No | 2GB | 4,876,405 | 428M |
| | OpenLegalData: German cases and laws | No | 5.4GB | 308,228 | 1B |
| | Charite doctoral theses abstracts | No | 28MB | 16,947 | 5M |
| | Flexikon | No | 106MB | 74,136 | 23M |
| | NTS of Animal Experiments | No | 24MB | 50,310 | 4M |
| | German Guideline Program in Oncology | No | 13MB | 4,348 | 3M |
| | Springer Abstract | No | 79MB | 34,035 | 15M |
| | CC medical texts (GER) | No | 3.6GB | 2,000,000 | 682M |
| | Medicine Dissertations | No | 1.4GB | 14,496 | 295M |
| | Pubmed abstracts | Yes | 8.5GB | 21,044,382 | 1.7B |
| | MIMIC III | Yes | 2.6GB | 24,221,834 | 695M |
| | PMC-Patients-ReCDS | Yes | 2.1GB | 1,743,344 | 414M |
| | German Fiction | No | 1.1GB | 3,219 | 243M |
| | English books | Yes | 7.1GB | 11,038 | 1.6B |
| | Total | | 167GB | 116,079,769 | 35.8B |

Table 1: Sizes of used data sources after pre-processing in the form of bytes, number of documents, and tokens. We integrate from top to bottom: formal, informal, legal, medical, and literature data.

domains, we strive to incorporate a wider range of linguistic styles and content. We sacrifice quality by adding automatically translated texts, filtered web-crawl data, and potentially noisier data from social media with the intention of increasing the data variety. To ensure that our results are comparable in terms of data volume, we take the step of removing a random subset of the GC4 dataset. We refer to this dataset as $\mathcal{D}_{\text{variety}}$.

**Informal**   Drawing inspiration from the work of Blombach et al., 2020, we have identified German content within the Pushshift Reddit Dataset (Baumgartner et al., 2020). To accomplish this, we gathered all submissions and comments posted between 2019 and 2023 from the dataset. We then performed several pre-processing steps, including unescaping HTML content, removing URLs, and filtering out texts with fewer than 20 words. To identify German content accurately, we employed the fasttext language identification model (Joulin et al., 2017). We set a threshold language identification score of 0.9, excluding posts that scored below this value.

As a result of this process, we obtained a collection of 15 million texts from Reddit. However, to augment our informal data further, we incorporated an additional 4.9 million texts from a corpus of reviews sourced from a German holiday booking website, as published by Guhr et al., 2020.

**Medical**   In contrast to English, which benefits from existing large medical text corpora like MIMIC (Johnson et al., 2016), the availability of such resources in German is limited due to strict privacy regulations. However, we have managed to collect a relatively small set of public medical datasets and websites. Using this data, we applied the 5-gram approach described in (Wenzek et al., 2020) to filter the latest release of the OSCAR corpus for medical texts. Out of the 207 million total documents in OSCAR, we focused on selecting the top 2 million documents where the 5-gram model had the lowest perplexity. Additionally, we have gathered 14,496 publicly available medicine dissertations written in German.

To further enrich this dataset, an extended dataset of six million PubMed abstracts, clinical notes from MIMIC III (Johnson et al., 2016) and PMC-Patients-ReCDS (Zhao et al., 2023) was translated into German. Prior to translation, the documents underwent a tokenization process using the Stanza library (Qi et al., 2020), where they were parsed into individual sentences. These sentences were then grouped into chunks, each constrained to a maximum of 128 tokens. The number of tokens was counted using the tokenizer provided with the translation model. This constraint was considered since it was observed that larger chunks of text resulted in a decrease in translation quality. For the translation phase, the Fairseq WMT'19 English to German model[3] was used. The model was configured with a context length of 156 and employed a beam search with a single beam, a setting chosen to optimize translation speed.

**Legal**   To cover the legal domain, the OpenLegalData corpus consisting of more than 250,000 German cases and laws was used (Ostendorff et al., 2020b).

**Literature**   We employed the Corpus of German-Language Fiction to conduct pre-training on German literature. This corpus comprises 2,735 German-language prose books from the Gutenberg-DE Edition, supplemented by an additional 484 books that were manually translated into German. Additionally, we applied the previously mentioned method to translate the BooksCorpus adding another 11,038 books.

The resulting datasets are summarized in Table 1.

## 4   Experiments and Results

### 4.1   Pre-training

Following (He et al., 2021), we pre-train DeBERTa$_{base}$, DeBERTa$_{large}$ and with the same configuration they used on both of our datasets and DeBERTa$_{xlarge}$ only on $\mathcal{D}_{\text{variety}}$. In the following, we will refer to the models as GeBERTa$^{Q}$ and GeBERTa$^{V}$. We train with an accumulated batch size of 2048 for 1M steps on the dynamic masked-language modeling task with the AdamW optimizer (Loshchilov and Hutter, 2019). We utilize DeepSpeed ZeRO optimization and the DeBERTa Hugging Face implementation. We train a sentencepiece tokenizer (Kudo and Richardson, 2018) on a random sample of 100M sentences of each dataset. For the base and large models, we set the vocabulary size to 50k tokens, while the vocabulary of the xlarge model has 128k tokens.

---

[3] https://huggingface.co/facebook/wmt19-en-de (last access: 21-06-2023)

| Model | Formal | | | Informal | | Medical | | | Literature | |
|---|---|---|---|---|---|---|---|---|---|---|
| | GE14 | GQuAD | | GE18 | TS | GGP | GRAS | JS | DROC | Avg |
| | | F1 | EM | | | | | | | |
| **GBERT**$_{\text{base}}$ | 87.10 | 72.19 | 55.07 | 51.27 | 72.34 | 78.17 | 62.90 | 77.18 | 88.03 | 73.65 |
| | ±0.12 | ±0.82 | ±1.39 | ±1.4 | ±0.48 | ±0.25 | ±0.01 | ±3.34 | ±0.20 | ±0.50 |
| **GELECTRA**$_{\text{base}}$ | 86.19 | 74.09 | 56.87 | 48.02 | 70.62 | 77.53 | 65.97 | 71.17 | 88.06 | 72.71 |
| | ±0.5 | ±0.70 | ±0.89 | ±1.80 | ±0.44 | ±0.11 | ±0.01 | ±2.94 | ±0.37 | ±0.66 |
| **GottBERT**$_{\text{base}}$ | 87.15 | 72.76 | 56.13 | 51.12 | 74.25 | **78.18** | 65.71 | 74.60 | 88.61 | 74.05 |
| | ±0.19 | ±0.378 | ±0.32 | ±1.20 | ±0.80 | ±0.11 | ±0.01 | ±4.75 | ±0.23 | ±0.51 |
| **GeBERTa**$_{\text{base}}^{\text{Q}}$ | 87.80 | 78.01 | 62.19 | 51.81 | 74.70 | 78.08 | 66.04 | 80.13 | 87.67 | 75.53 |
| | ±0.19 | ±1.12 | ±0.7 | ±1.53 | ±0.86 | ±0.16 | ±0.84 | ±5.23 | ±0.32 | ±0.32 |
| **GeBERTa**$_{\text{base}}^{\text{V}}$ | **88.06** | **78.54** | **62.06** | **53.16** | **74.83** | 78.13 | **68.37** | **81.85** | **89.14** | **76.51** |
| | ±0.22 | ±0.32 | ±0.55 | ±1.39 | ±0.36 | ±0.15 | ±1.11 | ±5.23 | ±0.32 | ±0.32 |

Table 2: F1-scores and standard deviation of the base models.

Table 4 presents the hyper-parameters used in the pre-training of the different GeBERTa models.

## 4.2 Downstream Tasks

We evaluate the existing German models and ours on a comprehensive benchmark. The benchmark encompassed various task types, including question-answering, classification, and named entity recognition (NER). Additionally, we introduced a new task focused on hate-speech detection, utilizing two existing datasets. When the datasets provided train, development, and test sets, we utilized them accordingly. In cases where such sets were not available, we randomly split the data into 80% for training, 10% validation, and 10% test set. Specifically for the GE18 dataset, we employed stratified splits based on the labels for the validation set.

**GermEval 2014 (GE14)** GermEval 2014 is a NER dataset based on German Wikipedia and news articles. It contains a total of 590,000 tokens from 31,000 sentences. It includes 12 entity types such as location and person.

**GermEval 2018 (GE18)** This multiclass hate-speech classification dataset comprises 5,009 German tweets. It contains two coarse and fine-grained annotations. We focus on the latter more challenging task. The dataset exhibits a significant imbalance, with only 11.9% of the samples marked as insults and a mere 1.4% labeled as profanity.

**Tweet Sentiment (TS)** We noticed that in GE18, the validation F1-score is not a good indicator of the test score. We believe this is due to the relatively small number of examples for the underrepresented classes. We, therefore, created another hate speech detection task from two existing datasets.

Guhr et al., 2020 introduced a large sentiment classification benchmark with positive, negative, and neutral labels. We combine sb10k (Cieliebak et al., 2017) and PotTS (Sidarenka, 2016), resulting in a dataset of 14,980 tweets.

**GermanQuAD (GQuAD)** For English models, the Stanford Question Answering Dataset (SQuAD) (Rajpurkar et al., 2016) is a popular downstream task. Each example consists of a paragraph from Wikipedia along with a question and the indices of the answer in the paragraph. In the second version, questions without an answer were added, increasing the task difficulty. Möller et al., 2021 created GermanQuAD, which is based on the German Wikipedia articles corresponding to the English ones used in SQuAD.

**GGPONC 2.0 (GGP)** Based on clinical practice guidelines in oncology (Borchert et al., 2022) published a NER dataset. Consisting of 1.8 million tokens from 10,000 documents, it is the largest available Germans medical NLP dataset. They published annotations with varying numbers of classes and span lengths. We focus on the most complex case with 8 fine-grained semantic classes and long entity spans.

**GRASCCO (GRAS)** is a collection of synthetic clinical case reports (Röhrig et al., 2022). Following the same annotation scheme as in GGPONC 2.0 an annotated version of GRASCCO was created (Bressem et al., 2023). Also, on this dataset, we evaluate the fine-grained classes and long entity spans.

**DROC** The Corpus of Character References in German Novels (DROC) comprises about 393,000 tokens from 90 German novels. 52,079 tokens were

annotated as character references differentiating between four entity types.

**Jsyncc (JS)** (Lohr et al., 2018) published a dataset of medical textbook cases with topic labels. Due to copyright restrictions, they released source code that generates the dataset based on the book PDFs. We were able to acquire 7 out of 10 of these books resulting in 544 samples and six classes.

## 4.3 Fine-Tuning

Following (Liu et al., 2019), we perform a limited parameter search for each model and task combination. We use a warmup step ratio of $6\%$ and consider the batch sizes 16 and 32. Depending on the model size we adapt the learning rate range. The maximum number of epochs was set to 20. The final parameters were set according to the best validation performance of each sweep. Finally, we perform five test runs with early stopping and report the average test performance.

## 4.4 Results

We present our results for the base models in Table 2. While GeBERTa$_{base}^{Q}$ surpasses the existing models on almost all tasks, GeBERTa$_{base}^{V}$ achieves the best results. Particularly clear differences compared to previous models can be seen on GQuAD ($+4.45\%$) and GRAS ($+2.4\%$). As expected, training on informal texts improves the results on downstream tasks from this domain, namely GE18, and TS. This is also true for the medical datasets and DROC. Interestingly, the performance also improves on GE14 and GQuAD. Both tasks are based on Wikipedia and can therefore clearly be assigned to the formal domain. Nevertheless, the model benefits from cross-domain pre-training here as well.

Table 3 presents the performance of the larger models. Analogous to the base models GeBERTa$_{large}^{Q}$ outperforms GeBERTa$_{large}^{V}$ although the difference here is smaller. For instance, on GQuAD it performs better than GeBERTa$_{large}^{V}$ ($+1.02\%$) and, interestingly, also on TS despite that it was not trained on informal data. This suggests that for larger models, the specific domains in the pre-training dataset are not always relevant. However, on GE18 it is clearly worse than GeBERTa$_{large}^{V}$. In addition, GeBERTa$_{large}^{V}$ achieves higher results on average across all tasks. Finally GeBERTa$_{xlarge}^{Q}$ scores the highest across all tasks but GGP and GRAS.

Our hyper-parameter search also improved the results that were previously reported on GE18 for GBERT$_{base}$ by $0.37\%$ and for GELECTRA$_{base}$ by $1.74\%$. Furthermore the performance of GottBERT$_{base}$ on GE14 by $0.31\%$. In some other cases, we were not able to reproduce the same performance as previously reported. We attribute this to the differences in evaluation. For instance, the results reported for GottBERT are the best result from several runs, while we report the average across five runs. Table 5 presents the hyper-parameter search space used for fine-tuning the models on the different downstream tasks.

The high standard deviation observed across all models in GE18 validates our initial concerns about its instability. Conversely, in the case of TS, the standard deviation is noticeably lower across all models.

## 5 Discussion

Our extensive experiments provide clear evidence that the models that we have published have achieved a significant step forward in the processing of natural language in the German language (NLP). It is important to note that certain improvements can undoubtedly be attributed to the DeBERTa architecture. This can be seen in improved performance on tasks such as question answering and named entity recognition with larger entity spans. However, a direct comparison between models trained on web crawl data alone and those trained on cross-domain data shows a clear improvement, which can be attributed to including diverse pre-training data. We support this claim with a significance test between GeBERTa$_{base}^{Q}$ and GeBERTa$_{base}^{V}$ which is shown in Table 6.

Furthermore, our results have shown the potential of a cross-domain dataset, mainly built by automatic translation and filtering of existing multilingual resources, to improve the quality of these models. The recently published Falcon model (Almazrouei et al., 2023) was trained using a combination of web-crawled data and curated English cross-domain data. Therefore, we strongly believe that generative models can also benefit from adopting our data collection methods for assembling multilingual, cross-domain datasets.

An interesting finding from our research is that our base model achieved a comparable level of performance to the previous large state-of-the-art models. This finding suggests that the required com-

| Model | Formal | | | Informal | | Medical | | | Literature | Avg |
|---|---|---|---|---|---|---|---|---|---|---|
| | GE14 | GQuAD | | GE18 | TS | GGP | GRAS | JS | DROC | |
| | | F1 | EM | | | | | | | |
| **GBERT**large | 88.48 ±0.23 | 81.51 ±0.84 | 63.41 ±1.01 | 54.37 ±1.65 | 73.60 ±0.61 | **79.17** ±0.14 | 69.28 ±0.80 | 76.32 ±4.42 | 90.29 ±0.15 | 76.63 ±0.63 |
| **GELECTRA**large | 88.39 ±0.13 | 80.51 ±0.41 | 63.71 ±1.15 | 55.41 ±1.54 | 73.84 ±0.86 | 79.09 ±0.09 | **70.16** ±0.92 | 73.73 ±2.35 | 89.83 ±0.27 | 76.37 ±0.69 |
| **GeBERTa**$^Q_{large}$ | 88.69 ±0.44 | 83.54 ±1.01 | 67.40 ±1.45 | 51.84 ±1.65 | 75.70 ±0.84 | 78.13 ±0.17 | 68.91 ±1.01 | 81.07 ±4.72 | 89.57 ±0.47 | 77.18 ±0.80 |
| **GeBERTa**$^V_{large}$ | 88.84 ±0.18 | 82.52 ±0.59 | 65.15 ±0.97 | 53.76 ±1.86 | 75.32 ±0.53 | 78.35 ±0.08 | 70.02 ±1.34 | 82.16 ±2.36 | 90.39 ±0.24 | 77.67 ±0.69 |
| **GeBERTa**$^V_{xlarge}$ | **89.04** ±0.26 | **85.05** ±0.63 | **67.71** ±0.62 | **55.80** ±1.42 | **76.25** ±0.704 | 76.71 ±0.08 | 67.92 ±1.00 | **82.42** ±4.70 | **90.63** ±0.21 | **77.98** ±0.62 |

Table 3: F1-scores and standard deviation of the large models.

putational resources can be significantly reduced by using a diverse dataset in conjunction with efficient model architectures. As a result, these advances become more accessible to a wider range of people and lead to savings in power consumption and a reduction in the associated environmental impact. A recently published study on a French model (Antoun et al., 2023) further supports this notion, showing that using novel model architectures helps to make training more data efficient. We hypothesize that training on diverse datasets could further improve efficiency in a similar way, given their training on a dataset derived from the CC100 pipeline.

## 6 Conclusion

In conclusion, this paper presents German language models that achieve state-of-the-art results on various natural language processing tasks by pre-training on more diverse datasets. The experiments show that training language models on cross-domain datasets leads to improved performance compared to models trained on web crawl data alone. The results suggest that the computational resources required for training can be significantly reduced while still achieving high performance through the use of diverse datasets and efficient model architectures. Overall, the research provides evidence that incorporating diverse, multilingual data is crucial for building high-quality language models.

## Limitations

Despite the remarkable capabilities of language models (LMs), it is important to recognise their limitations. One major concern is the high energy consumption associated with training LMs. The com-

putational resources required to train large-scale models can have a significant carbon footprint, contributing to environmental concerns.

Finally, while the majority of our dataset consists of filtered multilingual datasets and translations, this type of data is difficult to obtain in languages with limited resources. We believe that this is an issue that will need to be addressed in future work.

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

## A   Appendix

| Hyper-parameter | GeBERTa$_{\text{xlarge}}$ | GeBERTa$_{\text{large}}$ | GeBERTa$_{\text{base}}$ |
|---|---|---|---|
| Number of Layers | 24 | 24 | 12 |
| Hidden size | 1536 | 1024 | 768 |
| FNN inner hidden size | 6144 | 4096 | 3072 |
| Attention Heads | 24 | 16 | 12 |
| Attention Head size | 64 | 64 | 64 |
| Dropout | 0.1 | 0.1 | 0.1 |
| Warmup Steps | 10k | 10k | 10k |
| Learning Rates | 1e-4 | 1e-4 | 2e-4 |
| Batch Size | 2k | 2k | 2k |
| Weight Decay | 0.01 | 0.01 | 0.01 |
| Max Steps | 1M | 1M | 1M |
| Learning Rate Decay | Linear | Linear | Linear |
| Adam $\epsilon$ | 1e-6 | 1e-6 | 1e-6 |
| Adam $\beta_1$ | 0.9 | 0.9 | 0.9 |
| Adam $\beta_2$ | 0.999 | 0.999 | 0.999 |
| Gradient Clipping | 1.0 | 1.0 | 1.0 |

Table 4: Hyper-parameters for pre-training GeBERTa

| Hyper-parameter | xlarge | large | base |
|---|---|---|---|
| Warmup Ratio | 0.06 | 0.06 | 0.06 |
| Learning Rates | {5e-6,6e-6,7e-6,8e-6} | {7e-6,8e-6,9e-6,1e-5} | {1e-5,2e-5,3e-5,4e-5} |
| Batch Size | {16,32} | {16,32} | {16,32} |
| Weight Decay | 0.01 | 0.01 | 0.01 |
| Maximum Training Epochs | 20 | 20 | 20 |
| Learning Rate Decay | Linear | Linear | Linear |
| Adam $\epsilon$ | 1e-6 | 1e-6 | 1e-6 |
| Adam $\beta_1$ | 0.9 | 0.9 | 0.9 |
| Adam $\beta_2$ | 0.999 | 0.999 | 0.999 |
| Gradient Clipping | 1.0 | 1.0 | 1.0 |

Table 5: Hyper-parameter search space for different model sizes.

| Dataset | GE14 | GQuAD | GE18 | TS | GGP | GRAS | JS | DROC |
|---|---|---|---|---|---|---|---|---|
| p-value | 0.04 | <0.01 | 0.01 | 0.42 | 0.30 | 0.04 | 0.27 | <0.01 |

Table 6: t-test with the null hypothesis that the GeBERTa$_{base}^{V}$ achieves lower results than GeBERTa$_{base}^{Q}$. The test reveals statistical relevance (p-value < 0.05) for 5/8 datasets.