# OpenReview forum: "On the Impact of Cross-Domain Data on German Language Models"
_EMNLP/2023/Conference — EMNLP 2023 Findings_

### Official Review · Reviewer_BK21 · 2023-08-01

**Typos Grammar Style And Presentation Improvements:** N/A
**Soundness:** 3

**Excitement:**

2: Mediocre: This paper makes marginal contributions (vs non-contemporaneous work), so I would rather not see it in the conference.

**Missing References:**

N/A

**Paper Topic And Main Contributions:**

The author present a German dataset comprising texts from five domains, along with another dataset aimed at containing high-quality data.

**Questions For The Authors:**

N/A

**Reasons To Accept:**

The author build a German dataset comprising texts from five domains.

**Reasons To Reject:**

It looks like the authors merged multiple datasets and then processed them using other people's preprocessors, while using some automated programs to aid in the construction, but it's difficult to control the quality of the data when constructing with programs. I think the contribution of this paper is very limited.

**Reproducibility:**

5: Could easily reproduce the results.

**Reviewer Confidence:**

3: Pretty sure, but there's a chance I missed something. Although I have a good feel for this area in general, I did not carefully check the paper's details, e.g., the math, experimental design, or novelty.

---

> ### Author Rebuttal · Authors · 2023-08-29
>
> Thank you for your review. We want to highlight that our results show that cross-domain training outperforms quality-focused training, with up to 6.1% improvement over the previous state-of-the-art. Along with this result, we introduce (1) a German dataset comprising texts from five domains and (2) a number of pretrained large language models with state-of-the-art performance evaluated on a benchmark of 8 datasets. Our findings suggest that using a large corpus translated from English fills the gap of access to large-scale domain-specific German corpora, which will benefit NLP in many non-dominant languages.
>
> We appreciate your constructive feedback, and we are happy to address your concerns:
>
> **Dataset Composition**
>
> As you noted, we combined multiple datasets and employed a wide range of preprocessing methods. However, we would like to clarify that we did not rely on existing preprocessing scripts. Instead, we merged existing methods into a new framework for non-English variety-focused data acquisition and introduced new approaches, such as the translation of large English corpora and the filtering of common crawl for medical texts.  Since the translation models and the web-crawl data sets are available for a wide range of languages, we believe this work will also impact other languages. This finding holds relevance, particularly for languages where domain-specific resources are scarce.
>
> **Automated Programs and Data Quality**
>
> Your point about the challenges of maintaining data quality when employing automated programs is valid and is the core idea that drove our research. We believe there is a trade-off between data quality and dataset variety. While curated sources might restrict noise, incorporating diverse data, even from low-quality sources like social media, brings the advantage of capturing real-world variations. To investigate this tradeoff, we constructed two equally large datasets. The first dataset is geared towards data quality, incorporating carefully curated texts from reliable sources such as Wikipedia, news articles, and CC100 [1], for which the authors state that it consists of quality-filtered web data. In contrast, the second dataset showcases data processed using automated methods, including translation. We acknowledge that this data has lower quality due to automated processing. However, we show that despite this loss in quality, the variety-focused datasets lead to models that perform better. Beyond the specific nuances of data quality and variety, our work also introduces several pretrained large language models with state-of-the-art performance evaluated using 8 benchmark datasets for the German community.
>
> We appreciate your review and hope that we were able to address your concerns. We fully agree with your concern about the risk of low-quality data when applying automated processing scripts. Our results indicate that the benefits of data variety outweigh the drawbacks of lower data quality. We hope that our explanations provide a clearer perspective on the intentions and outcomes of our research. We are happy to respond to further questions and comments.
>
>
>
> [1] Wenzek, G., Lachaux, M.A., Conneau, A., Chaudhary, V., Guzman, F., Joulin, A., & Grave, E. (2020). CCNet: Extracting High Quality Monolingual Datasets from Web Crawl Data. In Proceedings of the Twelfth Language Resources and Evaluation Conference (pp. 4003–4012). European Language Resources Association.

---

### Official Review · Reviewer_Rbed · 2023-08-05

**Soundness:** 3

**Excitement:**

3: Ambivalent: It has merits (e.g., it reports state-of-the-art results, the idea is nice), but there are key weaknesses (e.g., it describes incremental work), and it can significantly benefit from another round of revision. However, I won't object to accepting it if my co-reviewers champion it.

**Paper Topic And Main Contributions:**

This paper describes the German corpus collected for training large German language models and analyses the performance of those language models. In the case of German, research on learning language models using large monolingual corpus is outdated, and unlike English, domain specific data is not actively used for learning. In this study, the authors first collected quality-focused data using CC100 and domain variety-enhanced dataset (adding informal, medical, legal, literature domain data to quality focused data).

In this paper, the following three models were trained by following the configuration of DeBERTa. First, $GeBERTa_{large}^Q$ is the $DeBERTa_{large}$ model trained with quality-focused data. $GeBERTa_{large}^{V}$ is a model with the same configuration trained on variety-enhanced data. Finally, $GeBERTa_{xlarge}^{V}$ is a larger model trained with variety-enhanced data.

The above models were finetuned on eight downstream tasks and their performance was compared with the models presented in previous studies. The results showed that the models trained with variety-enhanced data performed relatively better.

**Reasons To Accept:**

By releasing large amounts of German monolingual data for various domains and pre-trained models, which are not available in previous studies, it will be very helpful for future related research.

**Reasons To Reject:**

I don't see any reason to reject this paper.

**Reproducibility:**

4: Could mostly reproduce the results, but there may be some variation because of sample variance or minor variations in their interpretation of the protocol or method.

**Reviewer Confidence:**

4: Quite sure. I tried to check the important points carefully. It's unlikely, though conceivable, that I missed something that should affect my ratings.

---

> ### Author Rebuttal · Authors · 2023-08-29
>
> Thank you for your positive review. We appreciate your insights and the time you took for the review. We're pleased to hear your agreement with the potential impact of our work in the future, as well as the release of a number of pretrained large language models with state-of-the-art performance evaluated using 8 benchmark datasets, leading to improvements up to 6.1% over previous best results. We believe that these aspects will indeed prove beneficial for the broader community.
>
> Furthermore, we would like to take this opportunity to emphasize our contribution regarding empirical evidence for the benefits of variety-focused pretraining data as opposed to quality-focused data, particularly concerning encoder-only models. We're excited about the potential implications of our findings for the field, and we hope this contribution will resonate beyond the scope of German NLP.
>
> We are glad that you have recognized our experiments provide sufficient support regarding the advantages of variety-focused data over quality-focused data. We provide additional significance test results in our answer to reviewer 1.
>
> We thank you again for your positive assessment of our work, and we are happily available for any further questions that may arise.

---

### Official Review · Reviewer_oKki · 2023-08-05

**Soundness:** 3

**Excitement:**

3: Ambivalent: It has merits (e.g., it reports state-of-the-art results, the idea is nice), but there are key weaknesses (e.g., it describes incremental work), and it can significantly benefit from another round of revision. However, I won't object to accepting it if my co-reviewers champion it.

**Paper Topic And Main Contributions:**

This paper presents German language models that achieve strong results on several NLP tasks including GE14, GQuAD, GE18, TS, JS. The key finding of the paper is to claim that training language models on cross-domain datasets leads to improved performance compared to models trained on web crawl data alone.  Specifically, a large high-quality curated dataset from GC4, WMT 2022 German monolingual news crawl and Wikipedia is used as a baseline. Another dataset with the same size is used for comparison and that one includes cross-domain datasets related to Informal, Medical, Legal, Literature. Experiments show that training pretrained models with different size improve quality accuracy in 7/8 cases for base model and 6/8 cases for large model. While this is expected, it is not clear all of the improvement is significant. Another claim that the required computational resources can be significantly reduced by using a diverse dataset is misleading IMO though, because there is a lack of datapoints to convince this.

**Questions For The Authors:**

I noticed the batch size for training models is quite small (2048 sentences If I get it right). Is it expected?

[update after author response] I thank authors for their detailed response. I think the statement of the contribution from the paper should be updated as in your response. Please update the work that way. Meanwhile I update the score of the work from 2 to 3. I think it has merits but there are also key weakness (novelty, results are not that strong).

**Reasons To Accept:**

* Including cross-domain datasets seems to be useful in 7/8 cases and 6/8 cases. It is not obvious whether the gain is always significant (e.g. it seems very minor for GE14, GGP).
* Models will be released together with the paper and I think the released models will be useful for the community.

**Reasons To Reject:**

- The finding of having more cross-domain datasets for training pretrained models is not a novel idea  IMHO. It has been well known that the more diversed data we train a ML model, the better the model could be across different domains. With that to me the novelty of the paper is very limited.
- The claim that the required computational resources can be significantly reduced by using a diverse dataset is very misleading IMO. First of all if we train the a pretrained model on two different datasets (Q and V), not all improvements are that significant from their benchmark (maybe the biggest gains are with GE18, 68.37 and I am not sure the level of improvement is that significant). Second, it is true that the paper achieves much stronger results than previous work, but I think it is simply because the models trained with much larger data (Please correct me here if I was wrong - GBERT, GELECTRA are trained on GC4, while your models are trained on GC4 + WMT 2022 German monolingual news crawl + Wikipedia). This shows that increasing the training data is still the most crucial aspect regarding the training data and not the domains diversity). So I agree computational resources can be reduced, but I am not sure the level of degradation is that significant as claimed in the paper.

**Reproducibility:**

4: Could mostly reproduce the results, but there may be some variation because of sample variance or minor variations in their interpretation of the protocol or method.

**Reviewer Confidence:**

3: Pretty sure, but there's a chance I missed something. Although I have a good feel for this area in general, I did not carefully check the paper's details, e.g., the math, experimental design, or novelty.

---

> ### Author Rebuttal · Authors · 2023-08-29
>
> We thank you for your time, constructive feedback, and valuable questions raised.  We appreciate your sentiment that the models released with this paper will be of great benefit to the community. In the following, we love to address unclear aspects of the paper and answer insightful questions.
>
> **Addressing lack of contributions**
>
> The concept of cross-domain data is indeed not novel, and a positive effect of training on more diverse data is an intuitive hypothesis. However, we did not find conclusive evidence for this. For example, in the CamemBERT paper [1], OSCAR and CC100 were compared. The authors of CamemBERT reported no significant difference between models trained on OSCAR and CC100. While both are web-crawl-based, CC100 was filtered for quality content, potentially limiting its diversity. In contrast to this, the authors of “The Pile” [2] hypothesized that their improvement can be attributed to the higher variety in the pile compared to CC100. Most recently, in [3], the authors reported that carefully processed web data leads to better results than the pile. Supporting the claim that focusing on diversity outweighs focusing on quality.
>
> While evidence was shown for generative large language models we were not able to find any research on the effects of cross-domain training of encoder models. As this previous work led to no clear conclusion, we investigate the tradeoff between quality-focused data vs. variety-focused data for encoder-only models in a controlled setting where both datasets are equally large.
>
> Specifically, we would like to add that there is limited research on the effects of automatically translated English data cross-domain. For example, we found statistically significant improvements in the literature dataset DROC (see below), although almost all books used for pretraining were translations from English ones.  We also reported a significant improvement in the medical task GRAS (see below) while about 90% of our medical pretraining data was acquired through automatic translation or automatic filtering of common crawl. We think that this contribution is interesting for the non-English community, especially for low-resource languages.
>
> **Reduction of required computational resources**
>
> We agree that the performance improvement from the variety-focused data is not clearly visible for all downstream tasks. For clarification, we would like to note our claim of reduced computational resources also mentions the model architecture (i.e., DeBERTa) as a factor:
> “The results suggest that the computational resources required for training can be significantly reduced while still achieving high performance through the use of diverse datasets and efficient model architectures.“
> We support this claim with the observation that our base model with 122M parameters trained on V achieves an average F1 score of 76.51 while GBERT large scores 76.63, and GELECTRA large 76.37, both with 335M parameters, reducing the required model size almost by a factor of 3 while maintaining a similar performance. We attribute the fact that our base model performs similarly as large models to the variety-focused data, but also to the more efficient DeBERTa model architecture. Our base model trained on Q scored an average F1 score of 75.53, supporting the hypothesis that data variety is necessary to reach the performance of large models with base models and the reduction of computational resources that entails this.
>
>
> **Benchmark significance**
>
> Due to your interest in the significance of the results from the models trained on the Q and V datasets, we performed a t-test with the null hypothesis that the base model trained on V achieves lower results than the base model trained on Q.
>
>
> | Dataset | GE14   | GQuAD  | GE18   | TS     | GGP    | GRAS   | JS     | DROC   |
> |---------|--------|--------|--------|--------|--------|--------|--------|--------|
> |    p-value     | 0.04   | <0.01  | 0.01   | 0.42   | 0.30   | 0.04   | 0.27   | <0.01  |
>
>
> The test reveals statistical relevance (p-value < 0.05) for 5/8 datasets. Since the standard deviation for GE18 is much higher, we increased the number of runs initiated with random seeds from 5 to 10. Due to the limited available time for this rebuttal, we only performed additional runs for GE18.
>
> We would like to note that the statistically non-significant results are partly due to problems with the datasets. For example, the standard deviation in JS is very high, which can be explained by the very small size. In addition, GGP is the only dataset on which all models perform approximately equally well, which may indicate that this dataset might be saturated.
>
> **Correction of data quantity**
>
> While it is true that the size of the pretraining dataset plays a significant role in the quality of a model, GBERT and GELECTRA were trained on multiple datasets (see Table 1 in [4]) with a total size of 162GB. In comparison, our datasets consist of 167GB, which we only perceive as a minor difference in quantity and not as the primary factor for the performance difference.
>
>
> **Why this batch size**
>
> Each input consists of 512 tokens, so a batch size of 2048 means that each optimization step was done with 512 * 2048 tokens. We followed the original DeBERTa configuration [5].
>
> We sincerely appreciate your valuable feedback, which has greatly contributed to the refinement of our work. We hope that we addressed your concerns and will be happy to answer any further questions. If you think that our additional experiments and clarifications improve the quality of this work, we will happily include them in the final version.
>
>
> References
>
> [1] Martin, L., Muller, B., Ortiz Suarez, P., Dupont, Y., Romary, L., Clergerie, ., Seddah, D., & Sagot, B. (2020). CamemBERT: a Tasty French Language Model. In Proceedings of the 58th Annual Meeting of the Association for Computational Linguistics (pp. 7203–7219). Association for Computational Linguistics.
>
> [2] Gao, L., Biderman, S., Black, S., Golding, L., Hoppe, T., Foster, C., ... & Leahy, C. (2020). The pile: An 800gb dataset of diverse text for language modeling. arXiv preprint arXiv:2101.00027.
>
> [3] Penedo, G., Malartic, Q., Hesslow, D., Cojocaru, R., Cappelli, A., Alobeidli, H., ... & Launay, J. (2023). The RefinedWeb dataset for Falcon LLM: outperforming curated corpora with web data, and web data only. arXiv preprint arXiv:2306.01116.
>
> [4] Chan, B., Schweter, S., & Moller, T. (2020). German′s Next Language Model. In Proceedings of the 28th International Conference on Computational Linguistics (pp. 6788–6796). International Committee on Computational Linguistics.
>
> [5] He, P., Liu, X., Gao, J., & Chen, W. (2020). Deberta: Decoding-enhanced bert with disentangled attention. arXiv preprint arXiv:2006.03654.

---

### Meta-Review · Area_Chair_KybE · 2023-09-18

**Recommendation:** 4

**Metareview:**

This paper presents new sets of German encoder-only language models using a cross-domain dataset. The model is based on DeBERTa architecture, and it achieves an impressive performance on almost all the NLP tasks evaluated on compared to previous models (GBERT, GELECTRA, GottBERT).

The reviewers found the models development to be useful especially since it achieves impressive result and since the authors planned to release their code to the community. Also, their results highlight the use of a cross-domain/heterogeneous dataset for model pre-training. There is a small concern about limited contributions of the paper apart from the cross-domain corpus, and building a new set of models, but the issues have been addressed by the authors.

---

### Decision · Program_Chairs · 2023-10-07

**Decision:**

Accept-Findings

**Comment:**

This paper presents new sets of German encoder-only language models using a cross-domain dataset. The model is based on DeBERTa architecture, and it achieves an impressive performance on almost all the NLP tasks evaluated on compared to previous models (GBERT, GELECTRA, GottBERT).

The reviewers found the models development to be useful especially since it achieves impressive result and since the authors planned to release their code to the community. Also, their results highlight the use of a cross-domain/heterogeneous dataset for model pre-training. There is a small concern about limited contributions of the paper apart from the cross-domain corpus, and building a new set of models, but the issues have been addressed by the authors.